# Comprehensive Molecular Dissection of *Dermatophilus congolensis* Genome and First Observation of *tet*(Z) Tetracycline Resistance

**DOI:** 10.3390/ijms22137128

**Published:** 2021-07-01

**Authors:** Ian Branford, Shevaun Johnson, Aspinas Chapwanya, Samantha Zayas, Filip Boyen, Matylda Barbara Mielcarska, Lidia Szulc-Dąbrowska, Patrick Butaye, Felix Ngosa Toka

**Affiliations:** 1Ross University School of Veterinary Medicine, 42123 Basseterre, Saint Kitts and Nevis; IBranford@rossvet.edu.kn (I.B.); ShevaunJohnson@students.rossu.edu (S.J.); achapwanya@rossvet.edu.kn (A.C.); SamanthaZayas@students.rossu.edu (S.Z.); pbutaye@rossvet.edu.kn (P.B.); 2Department of Pathology, Bacteriology and Avian Diseases, Faculty of Veterinary Medicine, Ghent University, 9820 Merelbeke, Belgium; filip.boyen@ugent.be; 3Institute of Veterinary Medicine, Warsaw University of Life Sciences—SGGW, 02-786 Warsaw, Poland; matyldan@gmail.com (M.B.M.); lidia_szulc_dabrowska@sggw.edu.pl (L.S.-D.)

**Keywords:** *Dermatophilus congolensis*, genome, *tet*(Z), antimicrobial resistance AMR, virulence factors, secondary metabolites

## Abstract

*Dermatophilus congolensis* is a bacterial pathogen mostly of ruminant livestock in the tropics/subtropics and certain temperate climate areas. It causes dermatophilosis, a skin disease that threatens food security by lowering animal productivity and compromising animal health and welfare. Since it is a prevalent infection in ruminants, dermatophilosis warrants more research. There is limited understanding of its pathogenicity, and as such, there is no registered vaccine against *D. congolensis.* To better understanding the genomics of *D. congolensis*, the primary aim of this work was to investigate this bacterium using whole-genome sequencing and bioinformatic analysis. *D. congolensis* is a high GC member of the Actinobacteria and encodes approximately 2527 genes. It has an open pan-genome, contains many potential virulence factors, secondary metabolites and encodes at least 23 housekeeping genes associated with antimicrobial susceptibility mechanisms and some isolates have an acquired antimicrobial resistance gene. Our isolates contain a single CRISPR array Cas type IE with classical 8 Cas genes. Although the isolates originate from the same geographical location there is some genomic diversity among them. In conclusion, we present the first detailed genomic study on *D. congolensis*, including the first observation of *tet*(Z), a tetracycline resistance-conferring gene.

## 1. Introduction

The actinomycete *Dermatophilus congolensis (D. congolensis)* causes dermatophilosis, a disease mostly found in cattle, goats, sheep, and occasionally in horses [1]. This zoonotic disease is thought to be underdiagnosed in the human population [1]. In livestock, it is an economically important disease that hinders optimal animal productivity. Dermatophilosis is enzootic in the tropical and subtropical climates where the *Amblyomma variegatum* tick is the main transmitting vector. Animals of all ages can be infected and the disease severity ranges from mild to severe to lethal [2]. Clinically, dermatophilosis presents as both chronic dermatitis and systemic infections [3]. There is no targeted therapy for dermatophilosis in cattle, however topical or parenteral antibiotics have been used with success [3]. Tick control improves disease status [2] but no vaccine is currently available for dermatophilosis [4]. Given that differences in susceptibility of cattle have been associated with genetic markers, the selection of more resistant cattle has been beneficial in some countries [5].

*D. congolensis* has not been considered important, therefore, it has slowly become a neglected animal pathogen, that now is associated with the death of severely infected animals. Nevertheless, there is a lack of understanding of the molecular mechanisms involved in its pathogenicity. In addition, the genomic characteristics of the bacterium remain unclear, because only a few bacteria have been sequenced. Little is known about the important virulence factors [3] or potential antimicrobial resistance in this species, albeit a number of publications have indicated potential antibiotic resistance in this pathogen [1,6,7,8].

Due to paucity in comprehensive genomic information on *D. congolensis*, we sequenced 40 genomes from isolates of clinical bovine cases in St. Kitts where there was an outbreak causing sickness and death. Our results show that *D. congolensis* is a high GC member of the Actinobacteria whose genome encodes approximately 2527 genes, and among them, 23 housekeeping genes associated with antimicrobial susceptibility mechanisms and some isolates encode an acquired antimicrobial resistance gene, *tet*(Z), implicated in resistance to tetracycline. To the best of our knowledge, this is the first report on *D. congolensis* that encodes *tet*(Z).

## 2. Results

Out of 85 samples collected from animals with symptoms of dermatophilosis, 44 were cultured and successfully identified as *D. congolensis*. Identification by MALDI-TOF revealed score values between 1.72 and 2.25, which is in agreement with results reported by Alejo-Cancho et al. [9] regarding the identification of *D. congolensis.* Forty isolates were randomly selected for sequencing.

### 2.1. Genome Statistics

The 40 draft genomes were assembled from quality-filtered Illumina reads, in total 81,256,628. Taxonomic assignment checked in GTDB-Tk, showed that all the 40 isolates belonged taxonomically to *D. congolensis*. The 40 isolates of *D. congolensis* had a genome of approximately 2.6 Mb with a G+C content between 58.8% and 59.2%. On average, a *D. congolensis* genome contained 2527 coding sequences (CDS) (Table 1). There are 49 transfer RNAs (tRNA) present except for BTSK20, BTSK22, and BTSK31 which encode 52 tRNAs, while BTSK5, BTSK28, and BTSK34 encode 43, 46, and 48 tRNAs, respectively. All genomes have 3 ribosomal RNAs (rRNA).

Each of the 40 genomes has a single CRISPR array, but with various numbers of CRISPR spacers (Table 1) [10,11]. In four genomes, we detected a second CRISPR array, however, the evidence level was low, and we did not consider them for further analysis. The genomes have a Cas type IE and share 8 Cas genes (*Cas2_0_IE, Cas1_0_IE, Cas6_0_IE, Cas5_0_IE, Cas7_0_IE, Cse2_0_IE, Cse1_0_IE, Cas3_0_IE*) but different repeat consensus, where 2 genomes share GGCGGCCTGCCATCCATCCCCGGCGG, 17 genomes share GGCTCATCCCCGCAGGCGCGGGGAGCAC, and 21 genomes share the GTGCTCCCCGCGCCTGCGGGGATGAGCC sequence. Three different numbers of spacers were observed, 11 strains had 3–4, 20 strains had 5 and 9 strains had 26–31 spacers. The length of spacers was between 28–29 nucleotides. Other genome protection systems present are the restriction modification system in 12 genomes (*ApyPI* gene for a Type IIG restriction enzyme) and a methyltransferase with a recognition site at ATCGAC. This recognition site sequence found in the genomes has a nucleotide identity of 98% to the *ApyPI* gene. The mentioned genome protection systems were not identified in the reference genome NCTC13039 (Accession: GCA_900187045.1) or other available *D. congolensis* genomes in GenBank.

Two profiles of ANI were shown. For isolates BTSK1, BTSK2, BTSK7, BTSK8, BTSK17, BTSK21, BTSK27, BTSK31, and BTSK34, the ANI was on average 94.6% and for the remaining genomes it was >98%, thus within the range of ≥95%, which represents an accurate threshold for demarcating species in the Prokaryotes [12]. The representations of the identity distribution profiles are shown in Figure 1. The identity profiles are also supported by high bit scores. Bit scores measure sequence similarity independent of query sequence length and database size and are normalized based on the raw pairwise alignment score. The higher the bit score, the more highly significant the match is. This data was further supported by an additional analysis of 16s rRNA with SpeciesFinder v2.0 (https://cge.cbs.dtu.dk/services/SpeciesFinder/ (accessed on 5 February 2021)) in which all isolates were classified as belonging to *D. congolensis.* 16s rRNA sequences were also subjected to evolutionary analysis by the Maximum Likelihood method, which separated them into the same two groups as the ANI analysis (Appendix A).

### 2.2. Pan-Genome Analysis

The pan-genome of the 40 *D. congolensis* genomes revealed a total of 4665 CDS (Appendix A). The analysis showed that out of the 4665 CDS, 1775 CDS composed the core genome. With the average genome having 2527 CDS, approximately 70% of the genome accounts for the core genome in this collection of strains. The amino acid identity (AAI) calculated from the core genome of *D. congolensis* showed a 96.5% identity for 9 genomes and a 99.9% for the remaining 31 genomes, which agrees with the computed ANI percentages.

Further, we tested the concept of “open” and “closed” pan-genome based on the described mathematical method [14,15]. The prediction of the number of genomes needed to estimate the size of a pan-genome of the species was 1960 new CDS. This analysis was based on a reference genome and at least 15 draft genomes of *D. congolensis* (Figure 2A). Therefore, the pan-genome can be classified as open. The singleton development analysis predicted 37 CDS (Figure 2B) as a necessary addition to the pan-genome with each *D. congolensis* isolate sequenced.

### 2.3. Synteny Analysis

Synteny analysis can inform gene conservation among genomes. Although the 40 *D. congolensis* genomes appear to be quite similar, we sought to examine the possible evolutionary events in the genome arrangement. The EDGAR platform was used to compute the synteny matrices and plots. Figure 3 shows a synteny analysis of all 40 *D. congolensis* genomes with NCTC7915 as a reference and including 2 GenBank sequences of *D. congolensis*. The synteny indicates that the gene order is well conserved and supports the average amino acid identity (AAI) and ANI results. The only deflection to the gene order was represented by NCTC13039 and DSM 44,180 = NRBC 105,199 sequences, which are the only other *D. congolensis* sequences available in GenBank. Of the 40 isolates, only strain BTSK30 showed a discontinuity indicating a genomic variation at region 650–850 K. This suggests that the 40 *D. congolensis* from the West Indies are slightly different from those initially isolated from different geographical locations, i.e., DSM 44,180 = NRBC from Zambia (formerly Northern Rhodesia; NCTC13039).

### 2.4. Phylogenetic Analysis

We further compared the 40 isolates and constructed an SNP-based phylogenetic tree [16] (SNP Matrix Appendix A). The percentage of reference genome covered by all isolates was 84.25%. The tree is shown in Figure 4. The genomes were separated into three clusters. The first cluster encompassed genomes that originated from various locations on the Island. Cluster 2 was separated according to a single geographical location and 11 of the genomes in that cluster were positive for the *tet*(Z) gene. Cluster 3 comprised isolates from two other geographical locations and included a single isolate with *tet*(Z). Surprisingly, an additional feature inferred from the cluster separation was the number of CRISPR spacers, where cluster 1 had genomes with the highest number of spacers (27–33), cluster 2 had genomes with 6 spacers and cluster 3 genomes had 5–4 spacers. The map in Figure 5 shows the sampling areas.

### 2.5. Antimicrobial Resistance Mechanisms

Next, we investigated the presence of antimicrobial resistance genes in the studied *D. congolensis* genomes. Twelve isolates encoded the *tet*(Z) associated with resistance to tetracyclines [17]. The *tet*(Z) detected in the 12 genomes had an amino acid identity of 99.20% to that encoded by a closely related non-pathogenic bacterium *Corynebacterium glutamicum* 22243, in which this gene was originally described on a transposon carried by a plasmid [18]. *tet*(Z) has not been previously found in *D. congolensis*. Further, a total of 23 housekeeping genes associated with antimicrobial susceptibility mechanisms (Table 2 and Figure 6) were found in all isolates.

Analysis of *D. congolensis* with PlasmidFinder v2.1 (https://cge.cbs.dtu.dk/services/SpeciesFinder/ (accessed on 20 February 2021)) did not reveal any plasmids. We refined the analysis in PLACNETw to further address the question of plasmid presence in the genomes and found two *Rel* (Relaxase) genes associated with sequences similar to (1) pAG1 of *C. glutamicum* with 73.33% homology, and (2) a plasmid (*Gordonia westfalica*) MOBF with a homology of 45.42% (Figure 7). Upon further examination of the *D. congolensis* sequences in the vicinity of *tet*(Z), we found genes encoding a tetracycline repressor protein (TetR), similar to that described by Tauch et al. [18]. Two copies of a relaxase domain-containing protein, a type IV secretory system conjugative DNA transfer family protein, a site-specific recombinase, ParA-like protein, two copies of the replication initiation protein, DNA invertase, and two copies of excisionase family DNA binding protein were detected. The presence of these components indicates that the AMR gene *tet*(Z) is likely located on a plasmid.

### 2.6. Prophages

One prophage region was detected in isolates BTSK5, 6, 9–17, 24, 25, 27–30 at different locations on the chromosome, but having the same size, 8.9 kb. PHASTER classified them as incomplete prophages, which contained elements associated with phage structures, such as, base plate hub assembly protein, cell surface protein, membrane-associated initiation of head vertex, and resolvase. Detection of an incomplete prophage could probably be because no strictly unique bacteriophages have been described in *D. congolensis* and the sequences are likewise not recognized as phage related.

### 2.7. Virulence Factors

Further, we searched for potential virulence factors in the Virulence Factor Database (VFDB) [20,21]. Due to the absence of *D. congolensis* in VFDB, we ran our submitted pre-annotated draft genome in FASTA format against a distant relative bacterium, *Mycobacterium*. The detected potential virulence protein sequences in *D. congolensis* were then run through BLASTp to determine the identity of amino acid sequences to *Mycobacterium*. Since no pathogenic bacteria in the database are closely related to *D. congolensis,* a 50% identity was used to filter results obtained from the amino acid database. The VFDB analysis returned 19 potential virulence factor classes (VF) with a total of 49 genes encoding virulence factors (Appendix A).

The virulence factor representation was almost uniform in all the 40 genomes, with a few exceptions. In the Cell Surface Components VF class, *sugC* (a Trehalose-recycling-ABC transporter) was only present in half of the 40 genomes including the reference genome NCTC13039, and the other two genomes available in GenBank (NCTC7915 GCA_900447215.1; and DSM 44,180 = NBRC 105,199 NZ_AUCS00000000.1). In the Iron Uptake VF class, *fagA* (an ABC transporter) was present in the genome DSM 44,180 and not present in all the isolates. *sitB* (encoding a metal ABC transporter ATP-binding protein), apart from a few isolates was mostly absent including in the reference genome. A streptococcal plasmin receptor (*plr/gapA*) was identified in the Adherence VF class but was only identified in the NCTC13039 reference genome. Similarly, in the Endotoxin and Immune Evasion VF classes *cap5G* (capsule) and *epsE* (polysaccharide capsule), respectively, were found only in the reference genome. It should be emphasized here that the BLASTp analysis showed amino acid identity between 50–90% with most of the sequences between 50–60%, indicating the probability of functional and not structural similarity e.g., Gram-positive bacteria have lipoteichoic acid and not lipopolysaccharide. Further investigations should show functionality.

### 2.8. Metabolism and Secondary Metabolites

We employed BlastKOALA to predict functional gene clusters of the 40 isolates compared to the reference genome. Overall, 51% of the genes in the draft genomes fall into 21 functional clusters as analyzed by the Kyoto Encyclopaedia of Genes and Genomes orthology, compared to 54% of the reference genome (Appendix A). The little differences found in this comparison may indicate some flexibility of the genomes. We scrutinized the presence of non-ribosomal peptide synthetases (NRPS) in the 40 genomes of *D. congolensis* for virulence genes or antimicrobial resistance genes. Our analysis showed that all the genomes contained at least two NRPS clusters (Figure 8), with one cluster containing only 1 core biosynthetic gene and the other cluster containing both core and additional biosynthetic genes. In the first cluster (Figure 8A), the single-core biosynthetic gene encoded the non-ribosomal peptide synthase with an amino acid identity of 99.89% to the *D. congolensis* reference genome. The second cluster (Figure 8B) encoded the following, *MbtB* (phenyloxazoline synthase), thioesterase, saccharopine dehydrogenase NADP-binding domain-containing protein, *Gfo/Idh/MocA* family oxidoreductase, and amino acid adenylation domain-containing protein. We did not detect any resistance genes in both NRPS clusters. Detailed domain annotations for both NRPS gene clusters are shown in Figure 8 together with the predicted putative core structures of products. However, this is a prediction of the core structure of the molecule, and it is highly likely the final molecule is different. Sporadically, other secondary metabolite gene clusters were observed such as bacteriocins, siderophores, and terpenes (Appendix A). Further analysis in RAST (https://rast.nmpdr.org/ (accessed on 15 November 2020)) [22] revealed two more gene clusters for secondary metabolites, (1) Alkane synthesis cluster containing 4 genes; *OleA*—3-oxacyl-[ACP] synthase III, *OleB*—Haloalkane dehydrogenase-like protein, *OleC*—AMP-dependent synthetase/ligase, *OleD*—NAD(P) H steroid dehydrogenase-like protein, (2) Auxin biosynthesis cluster contained *APRT*—anthranilate phosphoribosyltransferase, *PRAI*—Phosphoribosylanthranilate isomerase, *Tsa*—Tryptophan synthase alpha chain, and *Tsb*—Tryptophan synthase beta chain.

## 3. Discussion

In this work, we provide a descriptive characterization of the main genomic components of *D. congolensis* isolated in a single geographical region. The data reported provides a valuable insight into the genome organization as well as the local variability and evolution. The de novo assembled genomes were compared to the existing reference and found not to differ significantly, at least as shown by analysis of the ANI. Computing ANI provides a better resolution of differences or similarities between the strains of the same species (80–100% ANI) [12]. In our data, this value was between 94.6% and 98.5%. This indicates also that the evolution of *D. congolensis* is rather slow because little variation is noticed. Thus, we assume that our observation indicates an evolutionary differentiation. However, a much larger difference would be expected if the bacteria were to evolve much faster. Arguably, this would be more accurately stated if strains from different time points were analyzed. In comparison to the reference genome, no differences were found in the GC composition (58.8–59.2%), but a slight difference in the overall number of CDS in the BTSK isolates and reference genome was observed, where BTSK isolates had an average of 2527 CDS and the reference genome NCTC13039 had 2267 CDS.

The few strains whose genomes are available in GenBank to appear to be relatively homogenous, as well as most of our strains. However, depending on local geographical distribution, we saw some more differentiation. Geographical location can impose different evolutionary forces that can lead to alterations in genome sequences. We minimized the number of passages after achieving pure cultures, so as not to exert unnecessary evolutionary forces. Therefore, the sequences obtained reflect the original field isolates well.

Interestingly, the phylogenetic tree inferred from the genomic data also separates the isolates into 3 clusters according to the three different numbers of CRISPR spacer sequences found in the genomes of isolates. The difference in number and length of CRISPR spacers have been used to differentiate Spiraeoideae-infecting strains of Erwinia amylovora from different geographical locations [24,25]. Although our analysis was not as elaborate, we observed that isolates in cluster 1 had two different sets of spacer sequences completely unique from those in cluster 2 and 3, while cluster 2 and 3 had two similar sets of spacer sequences, with the exception that each set of cluster 2 spacer sequences had an extra spacer sequence not found in cluster 3. The reasons why on such a small island, completely different spacers are found in the different isolates remain to be further investigated. A more geographically diverse collection of *D. congolensis* isolates will be necessary to better understand the diversity.

The synteny shows good conservation of the genome structure in our collection as well as compared to the reference strains. We obtained a typical X-alignment with 18 genomes having an inverted alignment which indicates the likelihood of genomic inversion occurring symmetrically at the origin of replication [26]. These results demonstrated little diversity among the strains indicating that the population is clonal, which can also be visualized by the phylogenetic tree of the strains. The limitation now is that the calculations are based on draft genomes, which may contain contig breaks that can influence CDS prediction and comparison.

It is known that mobile genetic elements (MGE) are critical to adaptive bacterial evolution and particularly play a critical role in the transfer of antibiotic resistance genes and virulence factors. We detected the *tet*(Z) gene and to our knowledge, this is the first observation of *tet*(Z) in *D. congolensis*. *tet*(Z) was first described by Tauch et al. [18] in C. glutamicum, but has been detected in other studies [27], and has been detected most recently in Rothia nasimurium [28]. It is thus a very uncommon gene, though present in the environment. It remains unclear what could be the origin of this gene on the island of St Kitts. The original description of *tet*(Z) located the gene on a plasmid (pAG1) [29], and in our strains it is most likely associated with a plasmid as well. The presence of this *tet*(Z) efflux pump in *D. congolensis* is worrisome because it can lead to therapeutic failure during treatment with tetracyclines, which are the recommended antibiotics for dermatophilosis. However, we have no data on the clinical implications of this resistance in the species and cannot assess whether it will negatively impair the treatment with tetracyclines.

Although we derived a considerable list of potential virulence factors found in the *D. congolensis* genomes, it remains a grey area as none of them have been functionally studied in this bacterium. However, assessing the protein identity to *D. congolensis* suggests the structural likelihood, ideally, the closely related species probably would have given a higher percentage of sequence similarity. Adherence is an important initial step in infection. We identified groEL with an amino acid identity of 72% to M. tuberculosis. It mediates the folding of proteins that are responsible for pathogenesis in M. tuberculosis. groEL knockout strains of *M. tuberculosis* were reported incapable of forming biofilms [30] and granulomas [31], suggesting groEL’s role in disease establishment or progression. Another gene associated with virulence was the sigA gene. It is an RNA polymerase sigma factor. Sigma factors are initiation factors that promote the attachment of RNA polymerase to specific initiation sites. sigA is a primary sigma factor in exponential bacterial growth [29]. According to Wu et al. [32], sigA modulates the expression of genes that contribute to *M. tuberculosis* virulence, enhancing the growth in human macrophages and in early phases of pulmonary infection in mice. It is possible that such a mechanism may be in play during *D. congolensis* infection since the infiltration of macrophages has been observed in skin lesions of dermatophilosis in cattle [33]. Overall, the sigma factors identified in *D. congolensis* genomes were significantly identical to those of M. tuberculosis, but, as mentioned earlier, unless experimental evidence is derived it is difficult even to speculate on their function in *D. congolensis*.

The genome of *D. congolensis* is in the smaller range, approximately 2.6 Mb, which may not allow for the encoding of many secondary metabolite gene clusters. The only well-represented gene cluster was the NRPS class as predicted by antiSMASH [23]. Rarely, the gene clusters such as bacteriocins, siderophores, and terpenes were found, however, their homology to similar proteins in other bacterial species was very low. This random occurrence of bacteriocins, siderophores, and terpenes may suggest their horizontal pattern of acquisition as reported for other bacterial species [34].

In conclusion, we provide a molecular characterization of the *D. congolensis* genome, which has a high GC content of 59.2% and encodes about 2527 genes. It has an open pan-genome, contains at least 49 potential virulence genes, and at least 23 antimicrobial resistance mechanisms, of which the *tet*(Z) is the only acquired resistance gene, and the first reported acquired resistance in this species. The analyzed draft genomes encode a single CRISPR array of Cas type IE with 8 Cas genes.

## 4. Material and Methods

### 4.1. Ethics Statement

The samples collected from animals were used for research, therefore, Institutional Animal Care and Use Committee approval was required (approval #18.03.08Toka). Animal owner consent for sample collection was also obtained.

### 4.2. Isolation of D. congolensis from Field Samples

Scab or swab samples (*n* = 85) were collected from cattle with dermatophilosis in different locations on the island of St. Kitts (17.36 N 62.78 W). Bacteria were isolated from scabs according to Quinn et al. [35] with a slight modification. Briefly, scabs were granulated in a sterile mortar with a pestle, and the resulting granular mass was placed in a tube containing 2 mL of distilled water and left to stand at room temperature for at least three hours. The tubes containing the fragmented scab material were later unsealed and placed in a glass jar containing CO_2_ emitted by a lit candle inside the jar for 15 min. The topmost content of the tube was collected and spread onto blood agar plates, which were then incubated in the presence of 5% CO_2_ for at least 72 h. Small, yellow, raised, beta-hemolytic colonies embedded in the agar were presumed to be *D. congolensis* and were stained with Giemsa to confirm the tramtrack morphology. Identification was confirmed by PCR [36] and Matrix-assisted laser desorption/ionization–time-of-flight (MALDI-TOF) [37].

### 4.3. DNA Isolation and Sequencing

Bacterial DNA was isolated with the AllPrep Bacterial DNA/RNA/Protein Kit (Qiagen, Calsbad, CA, USA) as described by the manufacturer. The quality and concentration of DNA were determined by Qubit 2.0. Illumina’s Nextera XT kit was used to prepare DNA libraries for sequencing. Admera Health, LLC, South Plainfield, NJ, USA was sourced for sequencing on Illumina HiSeq 2 × 150 platforms aiming at a sequencing depth of 30×. Quality control of reads was done with the FastQC software v0.11.5 (https://narrative.kbase.us) (accessed on 18 January 2020) [38]. Adapters and low-quality reads were removed by TrimGalore v0.6.4 (https://github.com/FelixKrueger/TrimGalore) (accessed on 5 February 2020) with parameters set as follows; quality 30: reads with Phred quality above 30; discard reads whose length is less than 50 bp after quality control; maintain paired-end reads order; retain reads that lost mate because of poor quality.

### 4.4. Genome Assembly, Annotation, and Analysis

The genomes were assembled with SPAdes v3.13.0 (https://narrative.kbase.us) (accessed on 20 February 2021), and the assembly quality was checked in QUAST v4.4 [39]. Taxonomic assignment was checked in GTDB-Tk v1.0.1 (https://narrative.kbase.us) (accessed on 30 June 2021) [40]. The genomes were annotated using the NCBI Prokaryotic Genome Annotation Pipeline v4.11 [41]. The Average Nucleotide Identity (ANI) was analyzed with the online tool by Environmental Microbial Genomics Laboratory (http://enve-omics.ce.gatech.edu/) (accessed on 1 March 2021) [13,42]. CRISPR arrays were searched with CRISPRCasFinder v 4.2.2 (https://crisprcas.i2bc.paris-saclay.fr/CrisprCasFinder/Index) (accessed on 2 March 2020) [10,11]. The Pan genome, core genome, and gene synteny were computed in EDGAR 3.0 (https://edgar.computational.bio.uni-giessen.de/cgi-bin/edgar.cgi) (accessed on 12 June 2020) [43]. The phylogenetic tree was constructed with CSIPhylogeny (https://cge.cbs.dtu.dk/services/CSIPhylogeny/) (accessed on 12 June 2020) as described by Kaas et al. [16].

We used the Comprehensive Antibiotic Resistance Database (RGI 5.1.0, CARD 3.0.7, https://card.mcmaster.ca/analyze/rgi) (accessed on 1 July 2020) [44] and ResFinder 3.2 (https://cge.cbs.dtu.dk/services/ResFinder/) (accessed on 1 July 2020) [45] to look for genes encoding antimicrobial resistance. PLACNETw (https://castillo.dicom.unican.es/upload/) (accessed on 1 July 2020), a web-graph-based tool for reconstruction of plasmids from next generation sequence pair-end datasets was used to search for the presence of plasmids [19,46]. PHASTER (BLAST+ v2.3.0+) https://phaster.ca/ (accessed on 15 August 2020) [47] identified prophage sequences.

Further, we searched for potential virulence genes using the Virulence Factor Database (VFDB) (http://www.mgc.ac.cn/cgi-bin/VFs/v5/main.cgi?func=VFanalyzer) (accessed on 19 January 2021) [20,21], which uses OrthoMCL to cluster the ortholog groups between representative strains and submitted strains, thus assigning a closely related ortholog. Due to the absence of *D. congolensis* in VFDB, we run our submitted pre-annotated draft genome in FASTA format against a distantly related bacterium Mycobacterium. Finally, we used antiSMASH (https://antismash.secondarymetabolites.org/#!/start) (accessed on 10 February 2021) [34] to examine the presence and characteristics of the secondary metabolite gene clusters within the *D. congolensis* genome. BlastKOALA https://www.kegg.jp/blastkoala/ (accessed on 5 May 2021) was used to determine the metabolic profiles of the isolates.

## Figures and Tables

**Figure 1 ijms-22-07128-f001:**
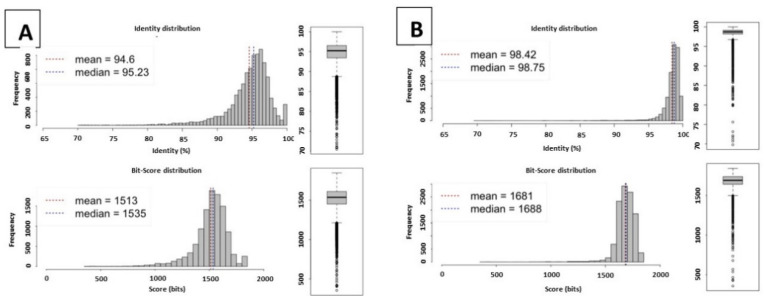
Estimation of the distribution of nucleotide identity between fragments of two genomic datasets. The estimation was conducted with an ANI calculator established by Kostas Lab according to calculations by Goris et al. [13]. (**A**) shows a representative of data on the 9 isolates with ANI at ≥94.6%; (**B**) shows a representation of data for the remaining 31 isolates with ANI at >98%.

**Figure 2 ijms-22-07128-f002:**
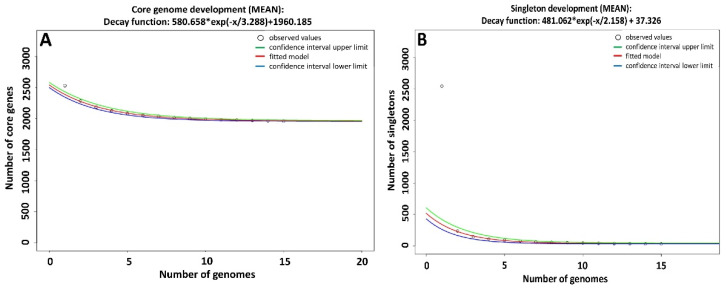
Singleton and Core-genome development plots calculated based on at least 15 representative *D. congolensis* genomes and a reference genome. It predicts that (**A**) the core genome size of *D. congolensis* is 1960 CDS and that, (**B**) each additional strain sequenced will add 37 new CDS to the pan-genome of *D. congolensis*.

**Figure 3 ijms-22-07128-f003:**
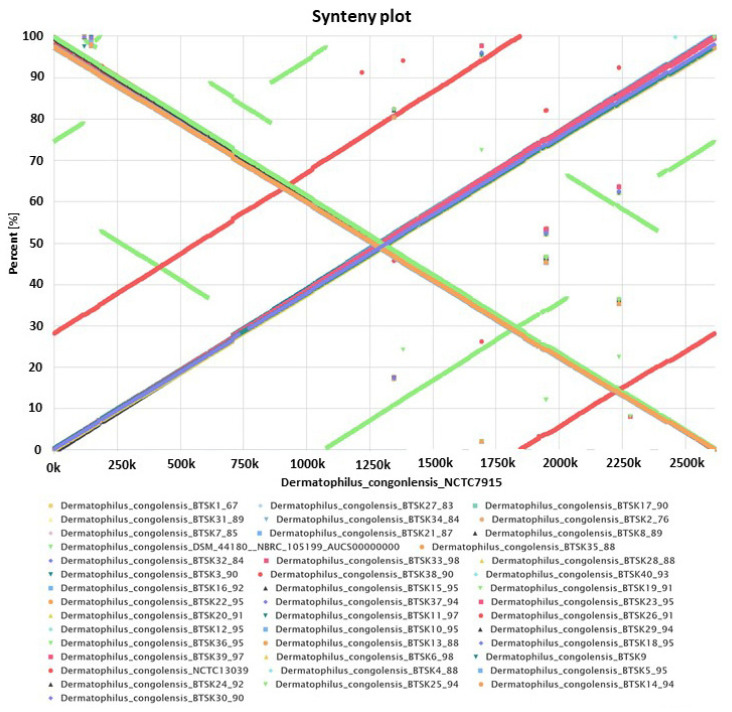
Synteny plot of 40 *D. congolensis* genomes depicting the conserved order of genes. We plotted all 40 because all 40 genomes in comparison to the reference showed the same gene order.

**Figure 4 ijms-22-07128-f004:**
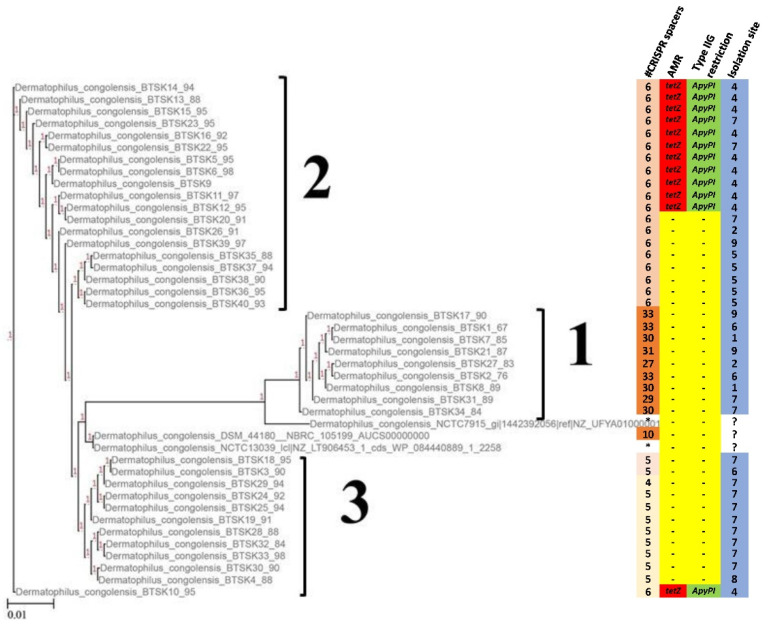
*D. congolensis* phylogeny. The phylogeny was inferred with the SNP calling procedure as described by Kaas et al. [16]. Three clusters are shown: cluster 1—genomes of strains isolated from different geographical locations on the island but having in common the higher number of CRISPR spacers (27–33); cluster 2—genomes of strains isolated in a single geographical location on the Island, and 11 of the genomes in the cluster are *tet*(Z) positive. All genomes in this cluster have 6 CRISPR spacers; cluster 3—genomes in cluster 3 of strains isolated in two geographical locations and have 5–4 CRISPR spacers. *—not detected, ?—origin not precisely described, “-“—not present.

**Figure 5 ijms-22-07128-f005:**
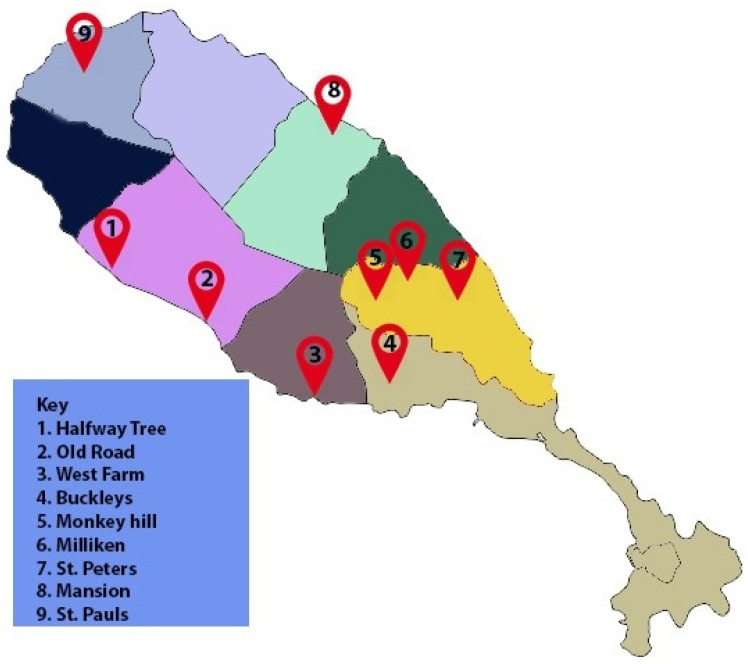
Map of St. Kitts showing the sampling areas.

**Figure 6 ijms-22-07128-f006:**
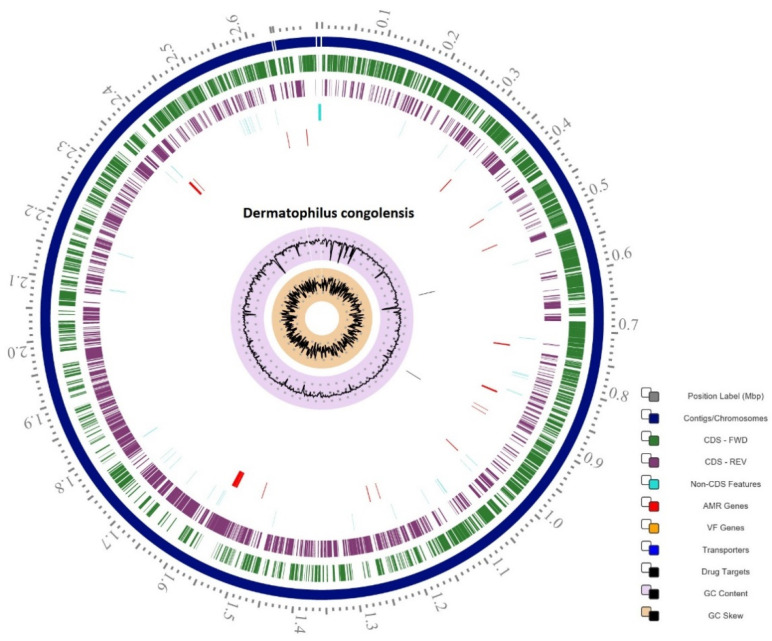
Circular view of a representative *D. congolensis* genome showing all features including antimicrobial resistance genes (red) and drug targets (black).

**Figure 7 ijms-22-07128-f007:**
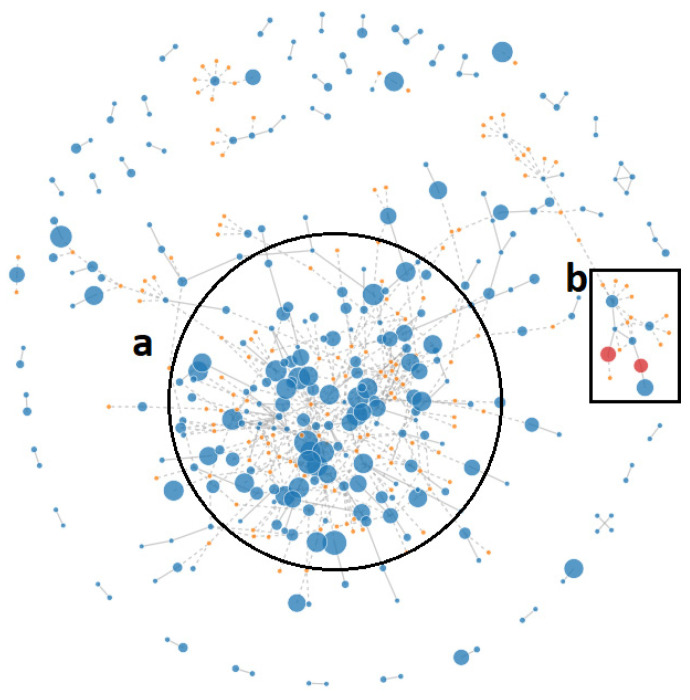
Unpruned, original view of *D. congolensis* genome in PLACNETw plasmid reconstruction software [19]. (**a**) Chromosome, and (**b**) putative relaxases associated with a pAG1 plasmid encoding TnpB, MyrA, *tet*(A), and *tet*(R). This analysis uses sequence paired-end reads. The orange nodes indicate a reference genome or plasmid; the blue nodes indicate a contig, and the difference in their size denotes the difference in length of contigs; the red nodes are contigs containing the Rel (relaxase) protein; dashed lines are scaffold links.

**Figure 8 ijms-22-07128-f008:**
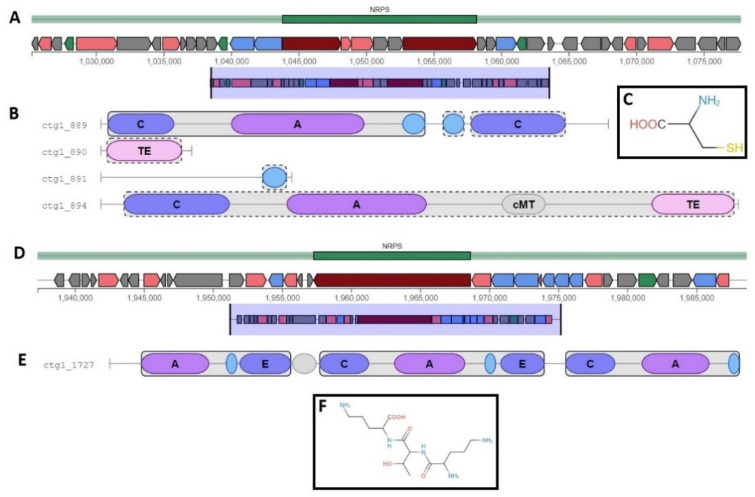
*D. congolensis* NRPS gene clusters, domain arrangements, and predicted core structure of products. Gene clusters, domains, and prediction of core structures were performed in antiSMASH [23]. (**A**) NPRS gene cluster containing two core biosynthetic and two additional biosynthetic genes. This cluster also contained other genes. (**B**) domain arrangement details C = Heterocyclization; A = AMP-binding with substrate prediction at consensus cys; cMT = Carbon methyltransferase; TE = Thioesterase. (**C**) the predicted core structure of the product molecule. (**D**) NPRS gene cluster containing a single gene. (**E**) domain arrangement; A = AMP-binding with substrate prediction consensus at orn; E = epimerization; C = condensation_DCL; A = AMP-binding with substrate prediction consensus at thr; E = epimerization; C = condensation_DCL; AMP-binding with substrate prediction consensus at orn. (**F**) predicted core structure of the product molecule.

**Table 1 ijms-22-07128-t001:** Summary statistics for 40 draft genome assemblies of *D. congolensis* isolated from acute cases of dermatophilosis on the Island of St. Christopher in the West Indies.

Isolate Name	Species	GenBank Accession No.	Genome Size (bp)	CDS	G+C (%)	AMR *	CRISPR-Array	CRISPR-Repeat	CRISPR-Spacer	Repeat-Region
BTSK1	*D. congolensis*	JAAFOV000000000	2,636,036	2482	58.8	22	1	33	32	4
BTSK2	*D. congolensis*	JAAFOU000000000	2,635,845	2501	58.8	22	1	33	32	4
BTSK3	*D. congolensis*	JAAFOT000000000	2,645,506	2506	59.2	22	1	5	4	2
BTSK4	*D. congolensis*	JAAFOS000000000	2,644,574	2504	59.2	22	1	5	4	2
BTSK5	*D. congolensis*	JAAFOR000000000	2,707,787	2566	59.2	23	1	6	5	2
BTSK6	*D. congolensis*	JAAFOQ000000000	2,707,101	2580	59.1	23	1	6	5	2
BTSK7	*D. congolensis*	JAAFOP000000000	2,634,952	2546	58.8	22	1	30	29	6
BTSK8	*D. congolensis*	JAAFOO000000000	2,610,909	2530	58.8	21	1	30	29	6
BTSK9	*D. congolensis*	JAAFON000000000	2,705,920	2577	59.1	23	1	6	5	2
BTSK10	*D. congolensis*	JAAFOM000000000	2,706,861	2581	59.1	23	1	6	5	2
BTSK11	*D. congolensis*	JAAFOL000000000	2,704,542	2564	59.1	23	1	6	5	2
BTSK12	*D. congolensis*	JAAFOK000000000	2,706,162	2567	59.1	23	1	6	5	2
BTSK13	*D.congolemsis*	JAAFOJ000000000	2,705,629	2562	59.1	23	1	6	5	2
BTSK14	*D. congolensis*	JAAFOI000000000	2,706,605	2520	59.1	23	1	6	5	2
BTSK15	*D. congolensis*	JAAFOH000000000	2,706,691	2553	59.1	23	1	6	5	2
BTSK16	*D. congolensis*	JAAFOG000000000	2,704,123	2561	59.1	23	1	6	5	2
BTSK17	*D. congolensis*	JAAFOF000000000	2,635,364	2551	58.8	22	1	33	32	4
BTSK18	*D. congolensis*	JAAFOE000000000	2,643,372	2486	59.2	22	1	5	4	2
BTSK19	*D. congolensis*	JAAFOD000000000	2,646,426	2505	59.2	22	1	5	4	2
BTSK20	*D. congolensis*	JAAFOC000000000	2,643,778	2498	59.2	22	1	6	5	2
BTSK21	*D. congolensis*	JAAFOB000000000	2,609,406	2513	58.8	21	1	31	30	4
BTSK22	*D. congolensis*	JAAFOA000000000	2,705,748	2584	59.1	23	1	6	5	2
BTSK23	*D. congolensis*	JAAFNZ000000000	2,707,702	2584	59.1	23	1	6	5	2
BTSK24	*D. congolensis*	JAAFNY000000000	2,644,395	2480	59.2	22	1	5	4	2
BTSK25	*D. congolensis*	JAAFNX000000000	2,644,659	2481	59.2	22	1	5	4	2
BTSK26	*D. congolensis*	JAAFNW000000000	2,690,948	2546	59.1	22	1	6	5	2
BTSK27	*D. congolensis*	JAAFNV000000000	2,635,575	2505	58.8	22	1	27	26	4
BTSK28	*D. congolensis*	JAAFNU000000000	2,644,675	2536	59.2	22	1	5	4	2
BTSK29	*D. congolensis*	JAAFNT000000000	2,641,543	2518	59.2	22	1	4	3	2
BTSK30	*D. congolensis*	JAAFNS000000000	2,692,568	2526	59.1	22	1	5	3	2
BTSK31	*D. congolensis*	JAAFNR000000000	2,635,532	2536	58.8	22	1	29	28	4
BTSK32	*D. congolensis*	JAAFNQ000000000	2,640,714	2499	59.2	22	1	5	4	4
BTSK33	*D. congolensis*	JAAFNP000000000	2,640,994	2490	59.2	22	1	5	4	4
BTSK34	*D. congolensis*	JAAFNO000000000	2,635,305	2527	58.8	22	1	30	29	4
BTSK35	*D. congolensis*	JAAFNN000000000	2,642,663	2488	59.2	22	1	6	5	2
BTSK36	*D. congolensis*	JAAFNM000000000	2,649,493	2496	59.2	22	1	6	5	2
BTSK37	*D. congolensis*	JAAFNL000000000	2,642,605	2517	59.2	22	1	6	5	2
BTSK38	*D. congolensis*	JAAFNK000000000	2,640,313	2503	59.2	22	1	6	5	2
BTSK39	*D. congolensis*	JAAFNJ000000000	2,640,884	2540	59.2	22	1	6	5	2
BTSK40	*D. congolensis*	JAAFNI000000000	2,640,858	2510	59.2	22	1	6	5	2

* Antimicrobial Resistance mechanisms.

**Table 2 ijms-22-07128-t002:** Antibiotic resistance mechanisms and associated genes detected in the 40 genomes of *D. congolensis* isolated from different locations on the Island of St. Christopher in the West Indies.

Gene	Product	
*rpoB*	DNA-directed RNA polymerase beta subunit (EC 2.7.7.6)	Antibiotic target in susceptible species. Rifamycins, Peptide antibiotics
*MtrA*	Two component system response regulator MtrA	Regulator modulating expression of antibiotic resistance genes. (azithromycin, erythromycin, penicillin)
*Ddl*	D-alanine--D-alanine ligase (EC 6.3.2.4)	Antibiotic target in susceptible species. Cycloserine
*S12p*	SSU ribosomal protein S12p (S23e)	Antibiotic target in susceptible species. Aminoglycosides (streptomycin)
*EF-Tu*	Translation elongation factor Tu	Antibiotic-resistant gene variant or mutant, elfamycin resistance gene
*rpoC*	DNA-directed RNA polymerase beta’ subunit (EC 2.7.7.6)	Antibiotic target in susceptible species. daptomycin
*Alr*	Alanine racemase (EC 5.1.1.1)	Antibiotic target in susceptible species. D-cycloserine
*OxyR*	Hydrogen peroxide-inducible genes activator = OxyR	Regulator modulating expression of antibiotic resistance genes. Isoniazid
*rho*	Transcription termination factor Rho	Antibiotic target in susceptible species. Bicyclomycins
*folA, Dfr*	Dihydrofolate reductase (EC 1.5.1.3)	Antibiotic target in susceptible species. Diaminopyrimidines: trimethoprim, brodimoprim, tetroxoprim, iclaprim
*PgsA*	CDP-diacylglycerol--glycerol-3-phosphate 3-phosphatidyltransferase (EC 2.7.8.5)	Protein altering cell wall charge conferring antibiotic resistance. Peptide antibiotics: daptomycin
*Iso-tRNA*	Isoleucyl-tRNA synthetase (EC 6.1.1.5)	Antibiotic target in susceptible species. Mupirocin
*gyrB*	DNA gyrase subunit B (EC 5.99.1.3)	Antibiotic target in susceptible species. Antibiotics Class: Fluoroquinolones Quinolones Quinolines, Aminocoumarin antibiotics
*folP*	Dihydropteroate synthase (EC 2.5.1.15)	Antibiotic target in susceptible species. Antibiotics Class: Sulfonamides
*gyrA*	DNA gyrase subunit A (EC 5.99.1.3)	Antibiotic target in susceptible species Antibiotics Class: Fluoroquinolones Quinolones Quinolines
*S10p*	SSU ribosomal protein S10p (S20e)	Antibiotic target in susceptible species Antibiotics Class: Tetracyclines, Glycylcyclines
*inhA, fabI*	Enoyl-[acyl-carrier-protein] reductase [NADH] (EC 1.3.1.9)	Antibiotic target in susceptible species Antibiotics Class: Isoniazid, Ethionamide, Triclosan
*dxr*	1-deoxy-D-xylulose 5-phosphate reductoisomerase (EC 1.1.1.267)	Antibiotic target in susceptible species Antibiotics Class: Fosmidomycin
*MtrB*	Two component system sensor histidine kinase MtrB	Regulator modulating expression of antibiotic resistance genes Antibiotics Class: Macrolides, Penams
*tet(Z)*	Tetracycline resistance, MFS efflux pump = *tet*(Z)	Efflux pump conferring antibiotic resistance Antibiotics Class: Tetracyclines
*gidB*	16S rRNA (guanine(527)-N(7))-methyltransferase (EC 2.1.1.170)	Gene conferring resistance via absence Antibiotics Class: Aminoglycosides
*EF-G*	Translation elongation factor G	Antibiotic target in susceptible species. Antibiotics Class: Fusidic acid
*kasA*	3-oxoacyl-[acyl-carrier-protein] synthase, KASII (EC 2.3.1.179)	Antibiotic target in susceptible species Antibiotics Class: Isoniazid, Triclosan
*rpsL*	SSU ribosomal protein S12p (S23e)	Aminoglycoside resistance gene, antibiotic-resistant gene variant or mutant

## Data Availability

GenBank accession numbers for genome sequences of each of the 40 isolates including their assembly statistics are given in Table 1 and were deposited in GenBank [48].

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
