# Peer review of "Comprehensive Molecular Dissection of Dermatophilus congolensis Genome and First Observation of tet(Z) Tetracycline Resistance"

_ijms, 2021, doi:10.3390/ijms22137128_

Round 1
Reviewer 1 Report
General Comments: The paper describes the genomic sequence of 40 isolates of the pathogenic D. congolensis bacterial species from the St. Kitts island. The authors identify numerous virulence factors, including the acquired antibiotic resistance gene tet(z), and genome preservation mechanisms contained in these isolates. The paper also indicates a high level of sequence similarity between the isolates, suggesting that this is a clonal population. Phylogenetic analysis reveals some sub-structure within this population. Overall a very solid, albeit standard description of bacterial genomic sequence. All appropriate analysis were done pertinent to a pathogenic bacterial species, and were described appropriately. A few interesting differences were identified between the strains, although for the most part these isolates were structurally very similar. Therefore there is no obvious merit in having all 40 sequences. A motivation for choosing this geographical location is missing and a description of the different sampling sites is needed. Is the clonal relationship of these pathogenic isolates expected in this geographical location? Do we know anything about the population size on this island? The authors state that their analysis reveals a ‘slow evolution’ rate for this species…but such a statement is only accurate if there was some aspect of time built into the study, and a discussion on the selective pressures on this organism, both of which are lacking. The reference strains used in this study show quite a deviation in synteny from the isolates, although the genomic sequences are significantly significant. Therefore a wider sampling of the species is needed to understand the genetic diversity, which the authors point out. Also, there is a difference in the number of CDS between the isolates and reference strain, but no analysis is done to understand functionality of these additional genes in the island isolates. Missing from the discussion is how this genetic analysis informs disease management. From a medical point of view, what have we learned? Specific Comments: Introduction needs more background/statements on the following topics: potential virulence factors in D. congolensis …what do we know/suspect already? lack of sequences in GenBank of closely related pathogenic species… this paper is filling a gap in knowledge Any significance to why this particular sampling site was chosen? Figure 1: comment on the Bit score in the results text, or remove these graphs. Figure 2 needs to be higher resolution Need to include an island map indicating the sampling sites and potentially highlighting the locations of the different clusters as defined by the phylogenetic analysis in figure 4 Phylogenetic tree can also be used to identify the branches on the tree with the different numbers of CRISPR spacers, the presence of Tet(Z) gene, the different genome protection mechanisms etc.. Mapping those genome/sequence features back onto the phylogenetic trees allows readers to see the molecular evolutionary trajectories of these strains, which would be interesting. line 47: tet(Z) gene is mentioned for the first time in the result section but it hasn’t been described yet. Recommend re-ordering the results section to bring section 2.5 on AMR forward and move the phylogenetic analysis later. Figure 5: Could not see any mark for loci identified as ‘transporters’ or ‘drug targets. either make the data more obvious, or remove those gene categories from the legend if none exist. Figure 6: legend incomplete - what do the colours represent? Why are some nodes different sizes. To a non-expert this is not obvious and needs to be detailed. Also, the figure appears to be cropped at the top. Please make sure all species names are italicised
Author Response
Dear Reviewer #1,
We thank the Reviewer for his effort and valuable time to review our manuscript. We have addressed all the concerns through a point-by-point response. We appreciate that the suggestions have improved our manuscript. Here we attach the responses in a separate file.
Thank you and Regards.
Corresponding Author

Reviewer 2 Report
In the present study, the authors determined 40 draft genome sequences of D. congolensis isolated from clinical bovine cases of dermatophilosis. The manuscript seems to be well written and the works may be performed with appropriate procedures. Although I think that the manuscript may satisfy the criteria for the publication, however, the manuscript still contains a few concerns for acceptance as follows:
- Does the presence of tet(Z) cluster have influence on the treatment of bovines suffered from dermatophilosis?
- The authors should emphasis advantages and significance on the observation of tet(Z) on D. congolensis with viewpoints of not only molecular biology but also clinical research in future.
Author Response
Dear Reviewer #2,
We thank the Reviewer for his valuable time and effort taken to review our manuscript. We have responded in a point-by-point format (please see attachment). We appreciate that the suggestions have improved our manuscript.
Thank you and regards.

Round 2
Reviewer 1 Report
Thank you for your thorough answers to the questions posied/comments made.